# OpenReview forum: "Don’t Say No: Jailbreaking LLM by Suppressing Refusal"
_ICLR.cc/2025/Conference — ICLR 2025 Conference Withdrawn Submission_

### Official Review · Reviewer_13TA · 2024-10-18

**Soundness:** 3
**Presentation:** 3
**Contribution:** 2
**Rating:** 5
**Confidence:** 3

**Summary:**

The paper explores jailbreaking attacks on LLMs, which prompt them to generate harmful content despite safety measures. It introduces the Don’t Say No attack, which improves attack success by suppressing the model’s refusals. To address challenges in evaluating such attacks, the authors propose an Ensemble Evaluation pipeline that uses Natural Language Inference and external LLMs for more accurate assessment. Experiments show the DSN attack’s effectiveness and the improved evaluation method’s accuracy.​⬤

**Strengths:**

Overall, the authors propose a new loss function with optimization tricks, the intuition "don't say no" is very intuitive.
- The idea is easy to follow.
- The exploration of token shift is interesting and supports the intuition in this paper.
- I appreciate the extensive and insightful experiments by the authors.
- The empirical results of this method seem promising.

**Weaknesses:**

I challenge the novelty of this paper comparing to GCG attack [1] and the analysis of jailbreak [2]:
- The main optimization procedure in this paper is modified from [1] with some previous wisdom borrowing from other papers, such as unlikelihood loss and cosine decay, which are mentioned in this paper.
- In addition, I believe that the algorithm in this paper is motivated by the jailbreak idea "Refusal Suppression" [2]. Therefore, someone may question the novelty of this paper as it seems like a combination of [1-2] and some previous tricks. I suggest the authors to provide a more detailed comparison of their method to these prior works, highlighting any key differences or improvements.

If the author's answers address my concerns, I will consider raising the score.

[1] Andy Zou, Zifan Wang, J Zico Kolter, and Matt Fredrikson. Universal and transferable adversarial attacks on aligned language models. arXiv preprint arXiv:2307.15043, 2023.

[2] A. Wei, N. Haghtalab, J. Steinhardt, Jailbroken: How does llm safety training fail? NeurIPS 2024.

**Questions:**

- The figure 1 looks a little blurry in some places.
- I'm attracted by the transferability of the method in this paper, its transferability performance seems much better than GCG [1]. Do you think this is caused by the nature of refusal suppression?

[1] Andy Zou, Zifan Wang, J Zico Kolter, and Matt Fredrikson. Universal and transferable adversarial attacks on aligned language models. arXiv preprint arXiv:2307.15043, 2023.

---

### Official Review · Reviewer_vdqn · 2024-10-27

**Soundness:** 2
**Presentation:** 3
**Contribution:** 2
**Rating:** 3
**Confidence:** 4

**Summary:**

Summary:  In this paper, the authors uncover why vanilla target loss is not optimal for jailbreaking LLMs, and then enhance the loss objective with the proposed DSN attack. The authors also target the evaluation challenge in studying jailbreak attacks such as the high false positives and false negatives, by proposing an Ensemble Evaluation pipeline.

**Strengths:**

+ proposed a new jailbreaking method via a suppressing refusal loss upon GCG
+ showed improved jailbreaking performances upon GCG

**Weaknesses:**

+ Technical novelty of the paper is somewhat limited and missing discussions with recent literature. The authors mainly change the loss of GCG attack by adding an extra refusal suppressing term. However, suppressing refusal is not something new. In fact, it has been mentioned in various previous works, e.g., [1][2]. Although the authors cited [1], they didn’t explicitly mention that refusal suppression is proposed there. [2] also talks about how to avoid refusal responses in decoding to jailbreak which shares a similar idea with the proposed work.

[1] Wei, Alexander, Nika Haghtalab, and Jacob Steinhardt. "Jailbroken: How does llm safety training fail?." Advances in Neural Information Processing Systems 36 (2024).

[2] Zhang, Hangfan, et al. "Jailbreak open-sourced large language models via enforced decoding." Proceedings of the 62nd Annual Meeting of the Association for Computational Linguistics (Volume 1: Long Papers). 2024.

+ The authors only compare with the traditional GCG attack. However, since GCG is proposed, many new attack baselines are presented. The authors might want to further compare with newer baselines (e.g. [3][4][5]) to demonstrate the effectiveness of the proposed work

[3] Chao, Patrick, et al. "Jailbreaking black box large language models in twenty queries." arXiv preprint arXiv:2310.08419 (2023).

[4] Li, Xirui, et al. "Drattack: Prompt decomposition and reconstruction makes powerful llm jailbreakers." arXiv preprint arXiv:2402.16914 (2024).

[5] Zhang, Tianrong, et al. "WordGame: Efficient & Effective LLM Jailbreak via Simultaneous Obfuscation in Query and Response." arXiv preprint arXiv:2405.14023 (2024).

+ It is also recommended to carefully check the attack performance under existing defense strategies, e.g., the following one.

[6] Cao, Bochuan, et al. "Defending against alignment-breaking attacks via robustly aligned llm." ACL 2024.

+ I appreciate the ensemble evaluation but I won’t count that as one of the major contributions of the work.

**Questions:**

+ Since the main argument is refusal suppression helps, I wonder if performance is related to the coverage of refusal statement diversity?

+ Can the authors be more specific on why the  Ensemble Evaluation is better than other mainstream evaluation strategies such as GPT4 evaluations or human evaluations?

---

### Official Review · Reviewer_a9Uq · 2024-10-31

**Soundness:** 2
**Presentation:** 3
**Contribution:** 2
**Rating:** 5
**Confidence:** 4

**Summary:**

Traditional optimization-based attack methods typically rely on an objective function designed to prompt the LLM into generating affirmative responses. However, these methods are limited by the fact that the affirmative responses considered during optimization are predefined. This paper addresses this limitation by introducing an additional loss function that suppresses refusal responses. The results demonstrate that incorporating this supplementary loss function can enhance the attack success rate. Furthermore, the paper contributes to the evaluation of jailbreak attacks by incorporating natural language inference (NLI) contradiction assessments, providing a more precise evaluation metric.

**Strengths:**

1.The approach of expanding the adversarial objective is highly relevant and addresses an important area of research.

2. The use of both GCG and AutoDan methods for evaluation, representing two typical types of adversarial suffixes (non-readable and readable), strengthens the results by demonstrating effectiveness across both categories.

**Weaknesses:**

1. The discussion about refusal words being narrower than affirmative responses is unclear and would benefit from further explanation.

2. The paper lacks concrete jailbreak examples for demonstration. Including cases where traditional loss functions fail to jailbreak LLMs, but the refusal-suppressing loss succeeds, would provide valuable insights and help clarify why suppressing refusals is effective.

3. In Section A.2.3, it is mentioned that seven refusal keywords are considered during training. An ablation study on the number or different combinations of these keywords would strengthen the analysis and provide deeper insights into their impact.

**Questions:**

1. Is there any arguments/experiments to show that refusal responses is narrower than affirmative responses?

2. Can you show some examples found by AutoDan + refusal suppressing loss?

---

### Official Review · Reviewer_qw9P · 2024-11-05

**Soundness:** 2
**Presentation:** 2
**Contribution:** 2
**Rating:** 5
**Confidence:** 3

**Summary:**

This paper studies LLM jailbreak attacks and proposes a refined optimization objective mainly based on the existing GCG methods. In particular, instead of aiming solely to generate affirmative responses, the proposed method introduces the DSN objective of suppressing refusal. Experiments show the effectiveness of the proposed method under white-box settings. Additionally, an ensemble evaluation pipeline is introduced to evaluate the success of jailbreak attacks more robustly.

**Strengths:**

+ The paper focuses on the timely problem of LLM jailbreak attacks.

+ The focus on learning-based jailbreak attacks is a plus, as it is different from most existing methods that rely on manually crafted prompts or templates.

+ The Experimental results are extensive, showing that the proposed method improves upon existing learning-based jailbreak attacks, such as GCG and AutoDAN.

**Weaknesses:**

- While the paper includes a set of variations compared with GCG, the proposed design is mostly based on intuitions. More in-depth analyses, either through more rigorous theoretical statements or well-designed experiments, are expected to support the design.

- The major limitations of GCG, such as limited transferability to black-box settings, high computational costs, and high perplexity of the optimized adversarial suffix, are not considered in this paper.

- The paper introduces concepts that are not defined clearly, making the paper difficult for readers to follow.

**Questions:**

I elaborate on my above comments on strengths and weaknesses below.

The paper focuses on a timely topic: improving the effectiveness of learning-based jailbreak attacks such as GCG. I appreciate the focus on the learning-based method, given that the majority of the literature in the field considers more heuristic ways of crafting jailbreak attacks. In addition, the proposed strategy of incorporating refusal suppression in the optimization objective is shown to be highly effective.

My main concern about the paper is that the proposed DSN attack heavily relies on the GCG optimizer and does not attempt to address its key limitations. While improving jailbreak attack success under white-box settings is meaningful to some extent, it is not very realistic for the attacker to access the gradient information to launch the attack. In the paper, there are not sufficient experimental results conducted for more black-box settings. Section 5.9 is fairly limited – I would expect more experiments testing the transferability of the proposed attacks to different target LLMs. I hope the authors can clarify in the rebuttal why they are focusing on white-box attack scenarios in the first place. Other limitations of GCG, such as high computation costs for optimizing the jailbreak suffix and high perplexity of the jailbreak prompts, are left unaddressed in the paper.

The concept of refusal suppression is not new; it has already been proposed in [1]. It is not surprising that incorporating a loss to suppress the refusal generation in GCG can improve the attack success rate. Section 3 provides some intuitions and empirical results as to why GCG fails, but they are not well-explained from my perspective. I could not understand the phenomenon of token shift and why incorporating an additional refusal suppression loss and cosine decay solves this issue. If the standard objective of GCG has issues, why still include the loss for generating affirmative response (instead of just using the refusal suppression term)? Can the authors provide more detailed explanations? I believe the authors are pointing to some general observations regarding jailbreak attacks aiming to generate affirmative responses, so more experiments documenting this phenomenon across various jailbreak methods and LLM models are expected.

In general, Sections 3 and 4 are difficult to follow. These sections introduce several vaguely defined terms, such as “vanilla target loss,” “token shift,” “true optimal solution,” “more constrained and predictable,” and “catastrophic loss term.” The authors should clarify those terms with mathematical definitions and avoid using vague terms throughout the paper.

Besides, some related work on learning-based jailbreak attacks, like [2], is missing, which should be discussed or even empirically compared in the paper.


[1] Jailbroken: How Does LLM Safety Training Fail? NeurIPS 2023

[2] AdvPrompter: Fast Adaptive Adversarial Prompting for LLMs: https://arxiv.org/pdf/2404.16873


__Some specific questions:__

1. How sensitive are the proposed method's attack success rates to the keyword list adopted in the training stage and the refusal matching metric? Why Cosine Decay weighting is defined as Equation (5)?

2. Equation 11 is difficult to follow. Why consider the metric of Shapley Value?

3. In Table 5, why is there a big gap between the ASRs achieved by $DSN_{mean}$ and that of $DSN_{max}$?

---

### Note · Authors · 2024-11-16

I have read and agree with the venue's withdrawal policy on behalf of myself and my co-authors.